# Factors associated with hypertension in Pakistan: A systematic review and meta-analysis

**Muhammad Riaz**[1,2]*, **Ghazala Shah**[3], **Muhammad Asif**[3], **Asma Shah**[4], **Kaustubh Adhikari**[2], **Amani Abu-Shaheen**[5]

1 Department of Public Health, College of Health Sciences, QU Health, Qatar University, Doha, Qatar, 2 School of Mathematics and Statistics, The Open University, Milton Keynes, United Kingdom, 3 Department of Statistics, University of Malakand, Lower Dir, Chakdara, Pakistan, 4 Department of Biotechnology, Women University Mardan, Mardan, Pakistan, 5 Research and Publication Center, King Fahad Medical City Riyadh, Riyadh, Saudi Arabia

* medical.stn@gmail.com

**Data Availability Statement:** All relevant data are within the manuscript and Supporting Information files.

**Funding:** The authors received no specific funding for this work.

## Abstract

### Background

High blood pressure is an important public health concern and the leading risk factor for global mortality and morbidity. To assess the implications of this condition, we aimed to review the existing literature and study the factors that are significantly associated with hypertension in the Pakistani population.

### Methods

We conducted several electronic searches in PubMed, ISI Web of Science, PsycINFO, EMBASE, Scopus, Elsevier, and manually searched the citations of published articles on hypertension from May 2019 to August 2019. We included all studies that examined factors associated with hypertension regardless of the study design. To assess the quality of the research, we used the Newcastle-Ottawa Quality Assessment Scale. We also conducted meta-analyses using the DerSimonian & Laird random-effects model to collate results from at least three studies.

### Results

We included 30 cross-sectional and 7 case-control studies (99,391 participants country-wide) in this review and found 13 (35.1%) to be high-quality studies. We identified 5 socio-demographic, 3 lifestyle, 3 health-related, and 4 psychological variables that were significantly associated with hypertension. Adults aged between 30–60 years who were married, living in urban areas with high incomes, used tobacco, had a family history of hypertension, and had comorbidities (overweight, obesity, diabetes, anxiety, stress, and anger management issues) were positively associated with hypertension. On the other hand, individuals having high education levels, normal physical activity, and unrestricted salt in their diet were negatively associated with hypertension.

**Competing interests:** The authors have declared that no competing interests exist.

## Conclusion

We found several socio-demographic, lifestyle, health-related, and psychological factors that were significantly (positively and negatively) associated with hypertension. Our findings may help physicians and public health workers to identify high-risk groups and recommend appropriate prevention strategies. Further research is warranted to investigate these factors rigorously and collate global evidence on the same.

## Introduction

Hypertension refers to a medical condition that presents with chronic elevation of systemic arterial pressure above a fixed threshold value. Several blood pressure (BP) thresholds have been established to date and the guidelines are regularly updated by several committees based on evidence from clinical trials. The American College of Cardiology and American Heart Association (ACC/AHA) updated their advice in 2017, as did the European Society of Cardiology and European Society of Hypertension (ESC/ESH) in 2018, followed by the National Institute for Health and Care Excellence (NICE) in 2019. All these agencies report to the Joint National Committee (JNC) and mostly publish similar recommendations, but they differ greatly in the threshold level of blood pressure for defining hypertension. The ESC/ESH defines hypertension at a BP greater than 140/90 mm Hg, while the ACC/AHA's most recent 2017 guidelines define hypertension at a lower threshold of BP >130/80 mm Hg [1,2].

Hypertension is a modifiable risk factor for many serious health conditions, such as peripheral vascular disease, end-stage renal disease, stroke, myocardial infarction, and congestive heart failure. It is the leading cause of mortality worldwide, accounting for at least 45% of deaths due to heart attack and 51% due to cerebrovascular incidents [3]. Uncontrolled hypertension leads to a greater risk of heart attack, heart failure, kidney diseases, stroke, lifelong disability, and death [4]. While pharmacological treatment effectively decreases the risk of coronary heart disease and other cardiovascular complications, non-pharmacological interventions, such as lifestyle changes, can also greatly reduce the morbidity and mortality of hypertensive patients [5].

Globally, about 1 billion people are living with hypertension as of today and according to the World Health Organization (WHO), this number expected to reach 1.56 billion people by 2025, covering 29.2% of the world's population [6]. The worldwide occurrence of hypertension among adults rose sharply from 594 million in 1975 to 1.13 billion in 2015 [7]. In developed countries, 30% of the adult population is hypertensive, and this number is expected to increase to 60% in the coming decades [8]. The worldwide distribution of hypertension is approximately 40.8% (Arab countries = 29.5%, Africa = 9.3–70.8%, Germany = 30.5%, and the United States = 30.5% in men and 28.5% in women) [9–12]. The prevalence of hypertension in Iran, India, China, and South Asian immigrants in the United Arab Emirates (UAE) is 41.8%, 29.8%, 29.6%, and 30.8%, respectively [13].

Hypertension is more common in low-income nations [14], where nearly 80% of deaths are caused due to cardiovascular diseases [15]. The prevalence of hypertension in Pakistan was reported by two large epidemiological studies: one study based on the National Health Survey (1990–1994) reported a rate of 19.1% [16] and the other based on the northern rural areas (2001) reported a rate of 14% [17]. According to another national health survey (2010), 33% of adults aged 45 years and 18% of all adults in Pakistan were hypertensive, and every third hypertensive aged 40 years and above was susceptible to an extensive range of diseases. This

survey also revealed that only half of the diagnosed hypertensive subjects were treated at any time, making the prevalence of controlled hypertension only 12.5% [18].

The risk of hypertension increases with several partially correctable lifestyle and dietary factors. According to WHO, a reduction in salt intake can decrease the risk of developing hypertension, stroke, and cardiovascular disease [19]. This may decrease other cardiovascular conditions since hypertension is linked with 47% of ischemic heart disease and 54% of stroke episodes universally [20]. In Gondar, Ethiopia, 66% of diabetic individuals were also found to be hypertensive [21].

In Pakistan, hypertension exists alongside or precludes chronic diseases, such as diabetes mellitus, cardiovascular disease, chronic kidney disease, and several behavioral and socio-demographic characteristics, including smoking, obesity, lack of exercise, and family history [22]. Although there is substantial research on the factors associated with hypertension in Pakistan, the findings of individual studies are not enough to inform clinical decisions. Therefore, systematic reviews valuable because they critically appraise all the available evidence and combine the results to reach a better conclusion about a research topic [23].

Therefore, we aimed to conduct a thorough systematic review and meta-analysis of the literature to make valid inferences about the risk factors of hypertension in Pakistan and provide greater consistency and validity to the previous results. We also highlighted the poor quality of existing literature on hypertension and the related methodological issues. We hope that our recommendations will help in the development of useful public health and clinical strategies to control hypertension and decrease the morbidity and mortality associated with it in Pakistan.

## Materials and methods

### Study area

This paper reviews studies from Pakistan, a developing country located in South Asia that covers an area of 881,913 km$^2$ (33$^{rd}$ largest country in the world). Pakistan has four provinces (Punjab, Sindh, Balochistan, and Khyber Pakhtunkhwa), one federal territory (Islamabad), and two autonomous territories (Gilgit-Baltistan and Azad Jammu and Kashmir). According to the latest census Figs, the total population of Pakistan was 197 million in 2017 (2.6% of the world's population), making it the sixth most populous country in the world. In 2012–14, the general government expenditure on health in Pakistan was only 1% of the GDP [24].

### Search strategy

Between May 2019 and August 2019, we searched the electronic databases of PubMed, EMBASE, PsycINFO, Elsevier, Scopus, and ISI Web of Science for all relevant and retrievable literature. The emphasis was on identifying all studies in Pakistan that investigated the factors associated with hypertension. We searched the following terms: "predictors", "factors", "determinants", "characteristics", "component", "psychological", "sociodemographic", "clinical", "health-related behavior", "health-related outcomes" (such as diabetes, liver disease, and heart disease), and family-related variables such as "father or sibling history of hypertension". Based on these terms, we created computer syntaxes for the online searches (S5 Table) and searched for combinations of the above-mentioned in the reference lists of retrieved articles. There was no restriction on language, but we found all the literature to be in English.

### Study selection

We included all studies that were conducted in Pakistan, reported factors associated with hypertension (self-reported or clinically diagnosed), and/or considered hypertension as the

main analysis outcome. There were no restrictions on the study design, age of the study population, and criteria for diagnosing hypertension. Studies that used statistical analysis to assess the association of various factors with hypertension, regardless of design, were included in this review. However, if a study was designed to assess a particular intervention and report it as a predictor of hypertension, it was not considered. Studies with narratives and case reports were also excluded.

We assessed all the identified studies for eligibility. Initially, reviewers G.S. and M.R. independently screened the identified studies by their titles first and then by the titles and abstract simultaneously. In the next step, they discussed the discrepancies and conducted a full-text review of the selected articles. Additionally, reviewer M.A. screened 2/3[rd] of the selected studies and extracted data independently. All the reviewers reached a consensus after addressing any disagreement or uncertainty in the study selection or data extraction processes.

## Data extraction

After screening the studies for inclusion, we developed a pre-designed data-extraction form to create a summary table of the important characteristics using the full text of the included studies. According to the form, G.S. initially performed data extraction under the supervision of M.R. The data was then verified by M.R. and M.A. (for 2/3[rd] of the studies). The data collection form retrieved information on the author's name, year and region (city) of the study cohort, aim and design of the study, patient characteristics (including age and anthropometric measurements), sample size and prevalence of hypertension, main outcome measure (hypertension), clinical validation of the outcome if applicable, factors associated with hypertension, statistical methods used, and effect estimates (e.g., odds ratios) for the factors significantly associated with hypertension. The process of screening and data extraction was carried out over 3 months (September 2019 to November 2019).

## Quality assessment of the studies

To assess the quality of the included research, we adopted the Newcastle Ottawa Quality Assessment Scale (NOQAS) developed to assess the quality of cohort studies and case-control studies [25] and its modified version that assesses cross-sectional studies [26,78]. The studies we included were either case-control or cross-sectional.

For case-control studies, the NOQAS is composed of three major components: a selection of the study groups (4 items), comparability of the groups (2 items), and ascertainment of exposure (3 items). Hence, by allocating one star per item, the maximum possible score could be 9 stars for one study. The Modified Newcastle Ottawa Scale (MNOQS) is composed of three main sections: selection of the study groups (4 items), comparability of the groups (2 items), and ascertainment of the outcome of interest (2 items). Hence, the maximum possible score for this scale could be 8 for one study.

Two reviewers (G.S. and M.R.) independently assessed the quality of the included studies and scored them. An MNOQS score of $\geq 7$ indicated a high-quality study; any study with a lower score was considered low-quality.

## Meta-analyses

Details on the statistical analyses used in the included studies are described briefly in the results section of this manuscript. We identified several factors associated with hypertension in Pakistan, and pooled the effect estimates, and quantified the amount of heterogeneity to obtain valuable information about the direction and size of effects for the association of a factor. We conducted the meta-analysis for a factor if its effect estimates from at least three studies

could be extracted or computed (based on the raw aggregated data available in the published manuscript). In all the included studies, the effect estimates (if available) were reported as odds ratios (ORs) 95% confidence intervals (95% CIs).

To ensure consistency of the various effect estimates in our meta-analysis, we changed the effect estimates for some factors so that we could use the appropriate reference category of the factor to assess its association with hypertension. For example, to perform a meta-analysis on older age as a categorical variable, we extracted or computed the ORs of hypertension for older age groups versus the youngest age group (<30 years). If a study reported an effect estimate for a continuous factor, we reported it separately.

Since the studies were mostly heterogeneous, we used the DerSimonian and Laird random-effects model to compute the pooled effect estimates and quantify the heterogeneity using the index $I^2$ [27]. This allowed us to compute the pooled effect estimates as OR (95% CI) along with $I^2$. We used the standardized normal statistic (z-score) to compute a p-value for testing the null hypothesis such that there was no effect of a factor on hypertension (effect estimate-OR = 1). A significance level of $p<0.05$ indicated a statistically significant association.

To adjust for multiple testing, we calculated the false discovery rate of controlling the multiple testing error rate using the Benjamini-Hochberg procedure [28]. We created a forest plot to display the results and used funnel plots to assess publication bias. To assess if the studies' quality affects the pooled results, we conducted sensitivity analyses by excluding low-quality studies while carrying out meta-analyses of factors reported by more than three good-quality studies (NOQAS>7).

This systematic review was completed and reported according to the Preferred Reporting Items for Systematic Reviews and Meta-Analyses (PRISMA) [29] (S4 Table).

## Results

### Literature search and study selection

Our primary search of the electronic databases returned 17,570 studies. We also manually identified 25 studies in the literature (using citations) that were not picked by the search as they were published before 1980. Thus, we identified a total of 17,595 studies for evaluation. However, 19 out of the 25 manually identified studies were not retrievable from online resources and some did not exist in the journal they were reported to have been published, as per the citations. On contacting the main authors or corresponding journals of these studies, we obtained only 8 out of 19 studies.

Initially, we identified 1,984 studies based on their titles. Of these, we excluded 1,919 after screening their abstracts. We then conducted a full-text review of 65 articles and further excluded 28 studies as they did not fulfill our inclusion criteria (Fig 1). Two reviewers (M.R. and G.S.) agreed (100%) to include the remaining 37 studies (7 case-control and 30 cross-sectional) in the present analysis.

### Characteristics of the included studies

Detailed characteristics of the included studies are described in S1 Table. We investigated 99,391 participants from 37 studies. All the studies were observational in nature; 7 (18.9%) were case-control [30–36] and 30 (81.1%) were cross-sectional. They were conducted in various regions across Pakistan and were published between 1980 and 2019. The sample sizes of the included studies ranged from 120 to 43,943 individuals. The participants' ages ranged from 5–80 years, and the study populations included medical students, university teachers, in-service/retired adults, businessmen, drivers/conductors, factory workers, clerks, shopkeepers, all working individuals, elderly individuals, and school children. Six studies were conducted in

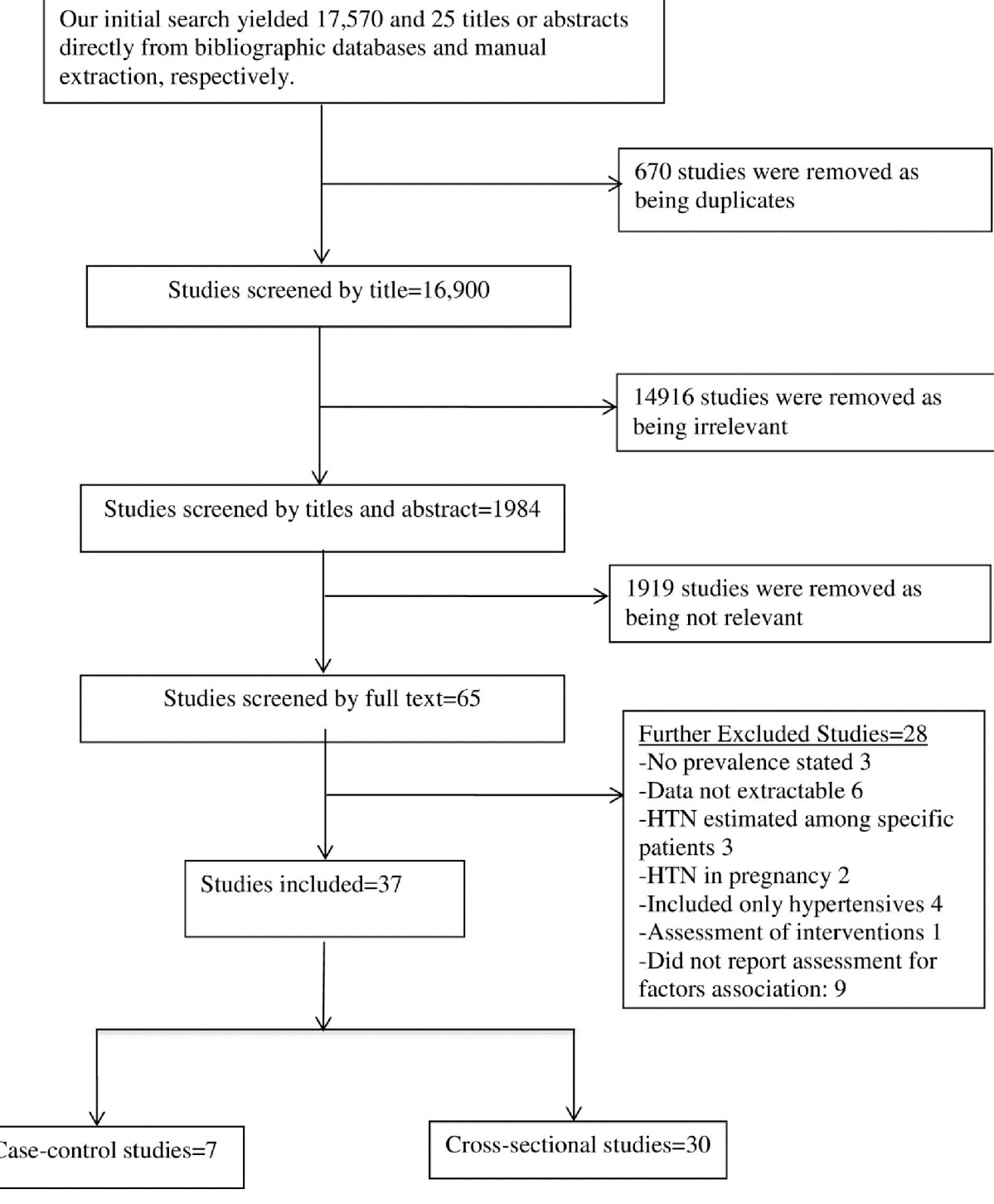

**Fig 1. Flow diagram showing the basis of inclusion and exclusion of studies in the review.**

rural areas [37–42], 8 were carried out in urban populations [43–50], and the remaining 23 (62.2%) sampled a combination of the two. Five (13.5%) studies included only women [37,51–54], 1 study [36] included only men, and 2 [48,50] were based on schoolchildren.

## Measurement of blood pressure and hypertension

In the included studies, BP was measured by trained investigators, attending physicians, community health workers, medical technicians, medical students, medical practitioners, and/or trained nurses using a manual mercury sphygmomanometer on the right arm with the patients in a sitting position. Three studies [39,42,55] employed an aneroid sphygmomanometer and 1 study [45] used the digital BP monitor OMRON. Out of 32 studies that described their BP measurement technique, 13 (40.6%) studies [31,38,39,42–44,52,56–61] did not report the frequency of BP readings, 8 (25.0%) [37,40,53–55,62–64] considered the average of two readings, 9 (28.1%) used the mean of three readings [36,41,45–48,51,65,66], and only 2 (6.3%) [30,50] used more than three BP readings. The initial rest time in all studies ranged from 2–5 minutes and the interval between two or more readings ranged from 2–20 minutes. All studies performed BP measurements in a single visit except 2 studies [30,66] that scheduled three and at least two visits, respectively.

Almost all the included studies followed the guidelines made by the Seventh Report of Joint National Committee (JNC-VII) [67] to diagnose hypertension (i.e., BP ≥ 140/90 or the use of medication for lowering BP), except for 2 studies [49,53]. In one study [49], self-reported hypertension or use of hypertensive medication was considered, and in the other [53], the reference range of BP to identify hypertension was not reported. Another study [46] used a cut-off value of BP ≥ 160/90 mm Hg for children aged 5–10 years and BP ≥ 165/90 mm Hg for adults aged 20–60 years. Four studies enrolled previously diagnosed cases of hypertension without specifying the standard methods used for diagnoses [32–35] (S1 Table).

## Study quality

Of 37 studies, 13 (35.1%; 7 cross-sectional: [40,41,45,52,57,61,66] and 7 case-control: [30,31,33–36,46]) scored at least 7 points out of 9 on NOQAS [25] and were considered high-quality. The median score was 6 (range: 4–8). All the case-control studies except one [30] and 7 cross-sectional studies were considered to be high-quality (score ≥7). The low-quality studies failed to control potential confounders in the design and statistical analysis, did not describe the sampling strategy, and lacked appropriate statistical methods (S2 and S3 Tables).

## Statistical methods used in the included studies

All the included studies contained statistical analyses that assessed the level of association between the outcome (hypertension) as a dependent variable and other factors as independent variables. Of the 37 studies, 17 (45.9%) studies (12 cross-sectional [30,36,39,41,42,45,50–52,57,58,61] and 5 case-control [31–35]) reported unadjusted ORs (95% CI) using logistic regression. Seven (18.9%) studies (5 cross-sectional [40,41,57,61,66] and 2 case-control [31,36]) reported adjusted ORs (95% CI) using multiple logistic regression analyses. Six (16.2%) studies (cross- sectional [38,43,46,55,58,64]) had descriptive analyses and the remaining 7 (18.9%) studies analyzed data using the Chi-Square test, t-test, or correlation analyses (S1 Table).

## Prevalence of hypertension

The overall prevalence of hypertension in the 37 included studies ranged from 3.0–77.5%. In cross-sectional studies and case-control studies, it ranged from 3.0–65.1% and 27.5–77.5%, respectively.

## Risk factors associated with hypertension

We extracted 46 factors that were significantly associated with hypertension from 37 studies conducted on the Pakistani population. There were 8 socio-demographic, 14 lifestyle, 14 health-related, and 10 psychological factors. The most frequently reported factors were: old age (15 studies), gender (14), urban vs. rural (4), level of education (5), physical activity (6), tobacco use (6), family history of hypertension (5), BMI categories (9), diabetes (6).

These and other less-frequently reported factors are presented in Table 1 along with their extracted estimates (95% CI), pooled effect estimates (95% CI) with p-values, and the heterogeneity index $I^2$. The meta-analysis results are displayed in Figs 2 and 3 and S1–S28 Figs. The sensitivity analyses (conducted by excluding low-quality studies) of several factors that were reported by more than three good-quality studies showed similar results with reduced heterogeneity (Table 1 and S28 Fig).

## Socio-demographic factors

The socio-demographic factors significantly associated with hypertension were: age in 16 studies [31,37–41,46–48,51,55,57,61,63–65], gender in 14 [38–41,46–49,57,60,61,63–65], income status in 3 [33,57,62], level of education in 5 [31,47,49,57,65], marital status in 3 [39,47,65], financial dependency in 1 study [49], and occupational noise level [66] and ethnicity (i.e., Muhajir, Pashtun, Baluchi vs. Sindhi) [57] in 1 study. The pooled effect estimates from our meta-analysis revealed that the risk of hypertension was increasingly significantly higher in

**Table 1. Factors significantly associated with hypertension: A meta-analysis.**

| Variables | Extracted Results:—study reference number: OR (95% CI)* | Pooled ORs (95% CI)[†], p, $I^2$ |
|---|---|---|
| **Socio-demographic and economic factors** | | |
| Age groups | Reference: [18–29] | |
| 30–39 | [30: 4.04 (1.54–10.61), 37: 5.13(2.83–9.27), 40: 3.48 (1.89–6.41), 41: 9.75 (3.06–31.08), 48: 1.21 (1.19–1.24), 59: 2.57 (1.37–4.82), 60: 2.18 (1.94–2.45)[#], 61: 2.76 (1.9–3.92)[#], 64: 1.01 (0.36–2.85)], | 2.65 (1.82–3.87), 2.3E-7, 95.4% 2.24 (1.71–2.95)[n], 46.8% |
| 40–49 | [30: 10.89 (2.85–41.64), 32a: 2.94 (1.92–4.52), 32b: 2.53 (2.06–3.11), 32c: 4.68 (4.00–5.47), 37: 74.68 (17.06–326.93), 40: 5.54 (2.95–10.43), 41: 19.07 (6.86–52.97), 45: 1.35 (1.01–1.81), 59: 5.75 (3.12–10.63), 60: 3.94 (3.51–4.41) [#], 61: 6.40 (4.56–8.99) [#], 64: 3.24 (1.31–8.03), 66: 2.92 (1.92–4.44) [#]] | 4.32 (3.24–5.76), 1.3E-23, 89.3% 4.12 (3.00–5.67)[n], 69.3% |
| 50–59 | [30: 21.39 (5.61–81.56), 32a: 4.89 (3.14–7.62), 32b: 3.88 (3.16–4.77), 32c: 10.87 (9.30–12.71), 33: 27.26 (14.59–50.94), 38: 1.77 (1.31–2.38), 40: 5.59 (2.90–10.75), 41: 37.56 (10.55–133.73), 45: 2.62 (1.96–3.48), 59: 13.31 (7.05–25.13), 60: 5.72 (5.03–6.50)[#], 61: 12.10 (8.55–17.12) [#], 64: 3.78 (1.51–9.49), 66: 5.60 (3.70–8.45) [#]] | 6.85 (4.77–9.85), 1.2E-25, 94.3% 6.63 (4.33–10.17)[n], 82.5% |
| 60+ | [30: 46.67 (5.03–433.09), 32a: 11.17 (6.75–18.48), 32b: 5.20 (4.31–6.28), 32c: 29.54 (25.71–33.94), 33: 10.89 (4.82–24.62), 38: 1.69 (1.13–2.54), 41: 26.96 (8.33–87.23), 45: 2.29 (1.76–2.98), 59: 17.53 (9.10–33.77), 60: 7 (6.08–8.05) [#], 61: 13.21 (8.78–19.86) [#], 64: 14.70 (3.69–58.63), 66: 11.49 (7.80–17.01) [#]] | 9.68 (5.49–17.10), 2.3E-15, 97.6% 10.16 (6.69–15.43)[n], 77.4% |
| Mean age (continuous) | [42: 1.21 (1.19–1.23)] [#] | |

*(Continued)*

**Table 1.** (Continued)

| Variables | Extracted Results:—study reference number: OR (95% CI)* | Pooled ORs (95% CI)†, p, I² |
|---|---|---|
| Gender (Male) | [32a: 0.98 (0.73–1.33), 32b: 1.23 (1.08–1.42), 32c: 1.06 (0.99–1.14), 33: 0.26 (0.16–0.41), 38: 0.81 (0.62–1.05), 40: 1.26 (0.71–2.25), 41: 0.30 (0.15–0.58), 42: 1.18 (1.04–1.35) #, 45: 1.18 (0.99–1.41), 54: 1.03 (0.58–1.82), 56: 2.00 (1.00–4.40), 58: 0.54 (0.32–0.91), 59: 1.66 (1.16–2.38), 60: 0.71 (0.59–0.84) #, 61: 0.87 (0.72–1.04) #, 66: 1.88 (1.48–2.37) #] | 0.97 (0.82–1.14),0.682, 87.9% 1.08 (0.75–1.54)[n], 49.5% |
| Marital status (married) | [33: 2.41 (1.49–3.92), 41: 2.02 (1.03–3.98), 59: 3.89 (2.19–6.90)] | 2.70 (1.88–3.88), 4.4E-8, 18.8%, |
| Urban residence (versus Rural) | [36: 2.16 (1.75–2.68) #, 42: 1.34 (1.20–1.49) # 42: 1.03 (0.89–1.19)#, 66: 3.03 (2.41–3.82)]# | 1.87 (1.27–2.76)[n], 0.002, 93.5% |
| Secondary or higher education versus (illiterate or primary education) | [33: 0.85 (0.55–1.29), 41: 0.23 (0.12–0.43), 42: 0.80 (0.69–0.93) #, 48: 0.54 (0.39–0.76) #, 58: 6.82 (3.85–12.08)] | 0.58 (0.38–0.88)[l], 0.011, 83.7%, |
| Income status: | | |
| Middle income vs. low | [31: 1.45 (0.58–3.65), 42: 1.07 (0.95–1.22)] # | 1.08 (0.95–1.22), 0.234, 0.0% |
| High income vs. low | [31: 1.73 (0.66–4.56), 42: 1.35 (1.15–1.57) #, 50: 1.23 (0.77–1.97) #] | 1.34 (1.16–1.55), 1.1E-4, 0.0% |
| Finically dependency | [58: 685.52 (173.15–2714.02)] | |
| Ethnicity: | | |
| Sindhi | 42: [1 (reference) | |
| Muhajir | 1.37 (1.10–1.69) | |
| Punjabi | 1.05 (0.88–1.27) | |
| Pashtun | 1.91(1.52–2.39) | |
| Baluchi | 2.71 (1.97–3.75)] # | |
| **Life style factors** | | |
| Physical activity (PA): | [33d: 0.03 (0.01–0.06), 33e: 0.09 (0.05–0.17), 35: 0.43 (0.26–0.72), 41d: 0.06 (0.02–0.15), 41e: 0.18 (0.09–0.37), 64e: 0.28 (0.13–0.62), 64d: 0.61 (0.31–1.23)] | 0.16 (0.07–0.35), 4.9E-6, 89.9%, |
| Unrestricted use of salt in the food | [33: 0.16 (0.10–0.26), 41: 0.20 (0.10–0.40), 61: 0.44 (0.26–0.72) #] | 0.24 (0.12–0.47), 1.5E-5, 76.4%, |
| High intake of meat | [42: 1.43 (1.24–1.64)] # | |
| Daily intake of Ghee/butter | [42: 0.74 (0.62–0.89)] # | |
| Daily intake of calcium (in food/milk) | [42: 0.86 (0.77–0.96)] # | |
| Housewives: (3 meals a day vs. no pattern) | [57: 2.38 (1.35–4.19)] | |
| having once a week fast food vs. sometimes (working women) | [57: 1.82 (1.11–2.98)] | |
| Working women | [57: 0.24 (0.17–0.34)] | |
| Housewives | [57: 4.13 (2.93–5.80)] | |
| Civil servants (male) | [40: 1.73 (1.19–2.50)] | |
| Factory workers (male) | [40: 0.56 (0.36–0.89)] | |
| Tobacco use | [35: 3.44 (2.24–5.30), 41: 0.27 (0.08–0.93), 42f: 1.00 (0.86–1.16)#, 42g: 1.38 (1.17–1.62)#, 47: 0.17 (0.07–0.44), 48: 1.56 (1.13–2.16)#, 61h:1.31 (1.06–1.61)#, 61i: 1.59 (1.31–1.92)#, 64: 1.30 (0.83–2.02)] | 1.48 (1.19–1.83)[m], 2.3E-4, 94.0%, 1.32 (1.12–1.56)[n], 71.6% |
| High level of occupational noise | [53: 3.01 (1.71–5.30)] | |
| Use of wine | [61: 1.47 (1.23–1.75)] # | |

(*Continued*)

**Table 1.** (Continued)

| Variables | Extracted Results:—study reference number: OR (95% CI)* | Pooled ORs (95% CI)[†], p, $I^2$ |
|---|---|---|
| **Health related variables:** | | |
| Family history of hypertension | [35: 2.63 (1.66–4.16), 48: 1.50 (1.07–2.11) [#], 60: 2.04 (1.80–2.30) [#], 61: 1.42 (1.14–1.76) [#], 64: 2.78 (1.76–4.40) [#]] | 1.91 (1.51–2.42), 4.2E-8, 72.9% 1.81 (1.40–2.34)[n], 76.4% |
| Body Mass Index: | | |
| Overweight | [38: 2.67 (1.74–4.10), 41: 2.02 (1.12–3.66), 47: 1.21 (1.09–1.34), 56: 2.60 (1.40–5.00), 60: 1.82 (1.67–1.98)[#], 61: 2.01 (1.51–2.67)[#], 64: 2.34 (1.46–3.75)[#], 66: 2.15 (1.51–3.07)[#]] | 1.95 (1.55–2.44), 4.5E-9, 86.7% 3.08 (2.60–3.64)[n], 18.8% |
| Obese | [33: 1.71 (1.09–2.69), 34: 2.11 (1.25–3.56), 35: 2.93 (1.91–4.50), 38: 6.61 (4.30–10.17), 47: 1.90 (1.76–2.04), 56: 4.30 (1.00–18.00), 60: 3.12 (2.84–3.43)[#], 61: 2.11 (1.30–3.40)[#], 62: 7.29 (2.26–23.55), 64: 4.71 (2.11–10.53)[#], 66: 3.38 (2.55–5.81) [#]] | 2.95 (2.26–3.84), 6.7E-16, 90.3%, 1.86 (1.72–2.01)[n], 0.0% |
| BMI continuous | [42: 1.04 (1.03–1.06)] [#] | |
| Diabetes | [37: 2.58 (1.09–6.09), 42: 1.37 (1.09–1.72) [#], 54: 5.38 (2.58–11.23), 58: 503.41(30.53–8300.22), 60: 1.95 (1.68–2.27) [#], 63: 9.72 (3.44–27.45)] | 2.94 (1.88–4.59), 1.1E-6, 85.4% |
| Abnormal fasting blood sugar (FBS) | [55: 2.16 (2.14–5.42)][#] | |
| Kidney disease | [48: 2.75 (1.80–4.20), 60: 1.85 (1.57–2.17)] [#] | |
| Headache | [44j: 5.56 (3.21–9.62), 44k: 0.18 (0.10–0.31)] | |
| Cardiovascular disease | [60: 2.98 (2.20–4.00), 37: 4.23 (1.48–12.08)] | |
| Lipid Abnormalities: | | |
| TC (Total cholesterol) | [46: 0.15 (0.06–0.39)] | |
| LDL-C (low density lipoprotein cholesterol) | [46: 2.67 (1.21–5.92)] | |
| HDL-C (high density lipoprotein cholesterol) in men | [46: 0.26 (0.10–0.67)] | |
| HDL-C in women (high density lipoprotein cholesterol) | [46: 0.09 (0.03–0.25)] | |
| TG (triglycerides) | [46: 0.03 (0.01–0.15)] | |
| Leptin(hyperleptinemia) | [55: 14.50 (5.01–35.60)] [#] | |
| Blood group O | [43: 5.05, p = 0.001] | |
| **Psychological factors:** | | |
| Anxiety | [49: 1.71 (1.29–1.40) [#], 50: 1.76 (1.09–2.85) [#], 52: 1.44 (0.72–2.19)][#] | 1.67 (1.34–2.09), 3.1E-6, 0.0% |
| Depression | [49: 1.64 (1.37–2.22) [#], 50: 1.44 (1.10–1.88) [#],] | |
| Stressful life (or Stress) | [49: 1.35 (1.12–1.62) [#], 50: 1.37 (1.01–1.85) [#], 52: 1.64 (0.86–2.13) [#], 64: 1.51 (0.84–2.72)] [#] | 1.39 (1.20–1.61), $3.7E^{-6}$, 0.0% |
| Components of Anger: | | |
| State-Anger | [49: 1.42 (1.25–1.66) [#], 51: 1.27 (1.09–1.47)] [#] | |
| Trait- Anger | [51: 1.16 (1.02–1.32)] [#] | |
| Anger Expression | [49: 5.73 (0.98–12.35) [#], 51: 1.85 (1.27–2.69)] [#] | |
| Anger-in | [49: 1.45 (1.30–1.70) [#], 51: 1.19 (1.06–1.33), 52: 1.73 (1.64–2.44)][#] | 1.42 (1.16–1.74), 3.4E-4, 83.1% |
| Anger-control | [49: 1.20 (1.11–1.30) [#], 51: 1.10 (1.00–1.21), 52: 1.20 (0.63–1.97)] [#] | 1.16 (1.09–1.23), 9.2E-7, 0.0% |
| Total Anger | [49: 1.08 (1.05–1.11)] [#] | |

(*Continued*)

**Table 1.** (Continued)

| Variables | Extracted Results:—study reference number: OR (95% CI)* | Pooled ORs (95% CI)†, p, I² |
|---|---|---|
| Psychological Distress (DASS) | [49: 1.4 (1.23–1.58)] # | |

This table includes extracted effect estimates for all factors (column 2) and pooled effect estimates (OR) for factors reported by at least three studies (column 3). A pooled OR>1 indicates positive association and <1 negative association of a factor with hypertension.

*: Most of the included studies did not report proper effect estimates (i.e., Odds Ratios) for the association of hypertension with the factors listed in the table. Therefore, data were extracted from the studies, and unadjusted odds ratios (ORs) and 95% confidence intervals (CI) for effect estimates were computed before conducting the meta-analyses.

†: Using DerSimonian and Laird random-effects model for meta-analysis, the effect estimates were combined to compute a pooled OR for the association.

#: Studies marked as of high quality by NOQAS, scoring at least 7 points.

p: is the p-value for testing the null hypothesis of (effect estimate (OR) = 1) versus an alternative of OR≠1). A conventional threshold of (p<0.05) shows that the effect is statistically significant. Applying the Benjamini-Hochberg procedure false discovery rate (FDR) method [28] on the 21 variables with meta-analyses, the corrected significance level is p≤0.011.

a: Metroville sample in study [63]: it is an urban community in the outskirts of Karachi Pakistan, a metropolis of over 15 million people.

b: Pakistan National Health Survey (PNHS) sample in study [63].

c: National Health and Nutrition Examination Survey in study [63].

d: In studies [47,55,65], effect estimate for "1–5 times/week PA (or average)" vs. sedentary.

e: In studies [47,55,65], effect estimate for "daily PA (or active)" vs. sedentary.

f: In study [57], the effect for "current cigarette smokers" was not significant but we have included in the Meta-Analysis (MA).

g: In study [57], the effect for "currently chew tobacco/snuff" was significant and was included in the MA.

h: In study [41], the effect estimate is for the "smoking".

i: In study [41], the effect estimate is for the "use of snuff".

j: Non-migraine or tension headaches.

k: Migraines.

l: In the meta-analysis for the level of education, the study [49] result being very odd, was excluded.

m: In the meta-analysis for "Tobacco use", the studies [47 and 65] results being very odd, were excluded.

n: Estimates obtained from the meta-analysis of high-quality studies.

older individuals (30–39, 40–49, 50–59 and ≥ 60 years) as compared with an age of <30 years and pooled ORs (95% CIs) of 2.65 (1.82–3.87), 4.32 (3.24–5.76), 6.85 (4.77–9.85) and 9.68 (5.49–17.10), respectively (Table 1, Fig 2, and S1 Fig). The risk was significantly higher among participants living in urban areas as compared with those living in rural areas with an OR of 1.87 (1.27–2.76). Married participants [2.70 (1.88–3.88)] and those with a higher income status [1.34 (1.16–1.55)] were more likely to be hypertensive.

On the other hand, the risk of hypertension was significantly lower in participants with a negatively associated socio-economic factor, i.e., a higher level of education (college and above) as compared with illiterate or primary education [0.58 (0.38–0.88)] (Table 1, Fig 2, and S2–S8 Figs).

## Lifestyle factors

We identified 3 lifestyle factors that were reported by at least three studies; they included physical activity (reported by 4 studies) [44,47,55,65], tobacco use (6 studies) [31,41,44,47,55,57],

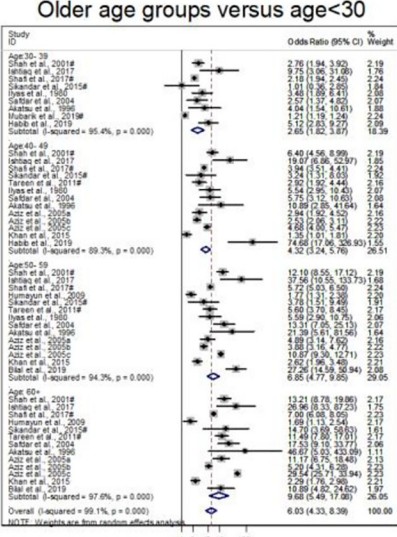

**Older age groups versus age<30**

#: Studies marked as of high quality by NOQAS, scoring at least 7 points.

The symbol ◊ and its size represents the pooled effect estimates-ORs (95% CI), and OR> 1

shows a higher risk of hypertension in overweight or obese compared with normal weight. The

heterogeneity coefficient (I-square= $I^2$) represents heterogeneity among the results; no

heterogeneity ($I^2$ =0.0%), very high (≥75.0%), moderate (50.0% <$I^2$ <75.0%), very low

($I^2$<25.0%).

**Fig 2. Forest plot from the meta-analysis (using the random effect model) of different age groups.**

and use of unrestricted salt in the diet (3 studies) [41,53,57]. Physical activity and unrestricted salt use were negative factors, i.e., they were significantly associated with a lower risk of hypertension, with pooled ORs (95% CI) of 0.16 (0.07–0.35) and 0.24 (0.12–0.47), respectively (Table 1 and S9–S12 Figs). On the other hand, tobacco use was significantly associated with a higher risk of hypertension, with an OR of 1.48 (1.19–1.83) (Table 1 and S13 and S14 Figs).

Only one study reported other diet- and work-related lifestyle factors, including a high intake of meat, daily intake of ghee/butter, and daily intake of calcium [57]; housewives who ate three meals a day vs. those who followed no eating pattern, working women who consumed fast food once a week vs. those who consumed it sometimes, and working women and housewives in general [53]; male civil-servants and factory-workers [46]; those working in an environment with a high level of occupational noise [66]; and consumption of wine [41]. Among these variables, the daily intake of ghee/butter, daily intake of calcium, working women, male factory workers, exposure to a high level of occupational noise, and consumption of wine were

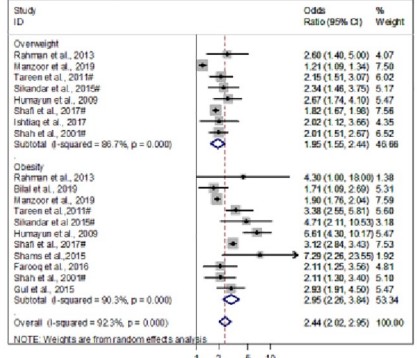

#: Studies marked as of high quality by NOQAS, scoring at least 7 points.

The symbol ◊ and its size represents the pooled effect estimates-ORs (95% CI), and OR> 1

shows a higher risk of hypertension in overweight or obese compared with normal weight. The

heterogeneity coefficient (I-square= $I^2$) represents heterogeneity among the results; no

heterogeneity ($I^2$ =0.0%), very high (≥75.0%), moderate (50.0% <$I^2$ <75.0%), very low

($I^2$<25.0%).

**Fig 3. Forest plot of the meta-analysis (using the random effect model) of different BMI groups.**

significantly associated with a lower risk of hypertension, and the other variables were significantly associated with a higher risk of hypertension (Table 1).

## Health-related factors

Three health-related factors were significantly related to hypertension and were reported by at least three studies were: a family history of hypertension [31,40,41,44,55], BMI (overweight and obesity vs. normal-weight) [36,40,41,43,44,47,48,52–55,57,59,61,64,65], and diabetes [36,40,42,49, 51,57,60]. The pooled ORs (95% CI) of the above factors were 1.91 (1.51–2.42), 1.95 (1.55–2.44), 2.95 (2.26–3.84) and 2.94 (1.88–4.59), respectively (Table 1, Fig 3 and S15–S20 Figs).

Other significantly associated but less-reported health-related factors were: a history of kidney disease [31,40], cardiovascular disease [40,51], dyslipidemia (TC, LDL-C, HDL-C, TG) [30], hyperleptinemia [36], abnormal fasting blood sugar [36], frequent headaches [58], and the blood group O [52]. The extracted effect estimates of these variables are provided in Table 1.

## Psychological factors

The psychological factors that were reported by at least three studies and were significantly associated with a higher risk of hypertension were: anxiety [32,33,35], stress [32,33,35,55], and

anger-in and/or anger-control issues [32,34,35]. The pooled ORs of these factors (95% CI) was 1.67 (1.34–2.09), 1.39 (1.20–1.61), 1.42 (1.16–1.74), and 1.16 (1.09–1.23), respectively (Table 1 and S21–S26 Figs). Additionally, depression [32,33], state-anger [34], trait-anger [34], anger-expression [34], total anger [34], and psychological distress [32] were reported to be significantly associated with a higher risk of hypertension (Table 1).

## Correction for multiple testing

Table 1 reports a meta-analysis of p-values for 21 variables. On applying the Benjamini-Hochberg procedure of false discovery rate [28] on the complete set of p-values, we found the corrected threshold of significance level to be p≤0.011, which was attained by 19 of the 21 p-values in our analysis (Table 1).

## Heterogeneity

The meta-analyses of all the factors that were reported by at least three studies revealed that there was a significantly high level of heterogeneity among all the studies in terms of the factors described, except for 4 (income status, anxiety, stress, and anger-control). The heterogeneity coefficient $I^2$ ranged from 0.0–97.6%, where no heterogeneity was indicated by $I^2 = 0.0\%$ for income status, anxiety, stress, and anger-control (Table 1, Figs 2 and 3, and S1–S3 Figs). The index was was very high (≥75.0%) for 12 factors (e.g., $I^2 = 97.6\%$ for the age groups of >60 years), moderate (50.0% $< I^2 <$ 75.0%) for a family history of hypertension and the age group of 30–39 years, and very low ($I^2 = 18.0\%$) for marital status. However, the meta-analysis of a factor with low or no heterogeneity included very few studies.

Heterogeneity among the studies may be due to differences in the study sample, sample size, study quality, use of different statistical methods (in some cases), BP measurement techniques, the definition of hypertension, measurement methods of a factor, and rates of hypertension observed. For example, by running meta-analyses for several factors based on good-quality studies, we observed that the heterogeneity index ($I^2$) among the pooled study had reduced considerably (Table 1, S27 Fig).

## Publication bias

The supplementary document contains the funnel and forest plots for each factor, which graphically represent the publication bias (S1–S27 Figs). Although the number of studies available for each factor is very small and it is difficult to form an opinion on the publication of each factor, the funnel plots for the results of some of the factors show evidence of some publication bias (S1–S3 and S10 Figs).

## Discussion

Systematic reviews are considered the gold standard in evidence-based medical practice and research. The present review and meta-analysis investigated factors associated with hypertension in Pakistan by including 37 studies on over 99,000 individuals. Based on the results of our meta-analysis, we extracted 46 factors that were significantly associated with hypertension. Of these, 9 factors were more frequently reported (in >3 studies), which included age, gender, urban residence, level of education, physical activity, tobacco use, family history of hypertension, BMI categories, and diabetes. Seven other significant but less-frequently reported (in 3 studies) factors included marital status, income status, unrestricted use of salt in the diet, anxiety, stress, anger in, and anger control. The remaining 30 factors (Table 1) were significantly associated with hypertension, but they were assessed by only one or two studies. We classified

all the factors into 4 major categories: socio-demographic and economic, lifestyles, health-related, and psychological.

Our findings will be important for developing public health policies in Pakistan and the world, and to drive interventions to prevent or treat hypertension, a leading cause of death and disease [68]. There are two strengths to our study. Firstly, the pooled evidence of all the factors investigated in this study can help clinicians and researchers identify individuals that require treatment and design interventions for those who are at a higher risk of hypertension; these may include older patients, those with a higher income, a family history of hypertension, those who are overweight and obese and lead a sedentary lifestyle, and those who habitually using tobacco. Some of these factors can be corrected and managed to prevent the onset of hypertension or treat it by controlling BP [20,21]. Older and overweight or obese patients are required to carry out moderate-to-vigorous physical activities and manage their diets daily to prevent or manage hypertension. Patients with psychological illnesses are also at a higher risk of hypertension and require special attention. Secondly, some health conditions may be directly caused by hypertension or could lead to hypertension. In a hypertensive state, there is an imbalance between vasoconstrictors and vasodilators that causes a change in the vascular tone and the fibrinolytic and coagulation pathways. This eventually causes organ damage, such as kidney failure and coronary artery disease [69]. Situational stress also temporarily raises the BP; stress hormones (adrenaline and cortisol) are released into the blood to prepare the body for "fight or flight" by increasing the heartbeat and constricting the blood vessels [69,70].

Some factors we reported to be associated with hypertension are similar to those reported in the global literature; they include older age, BMI (overweight or obesity), marital status, smoking or chewing tobacco, a sedentary lifestyle, family history of hypertension, and diabetes [70–77]. Additionally, we identified some other factors that are not well-studied globally and have been reported by only one or two studies in Pakistan; they include a high intake of meat, ghee/butter, and calcium, housewives and working women with unhealthy dietary habits, male civil-servants and factory-workers, and mental health aspects, such as depression, anger, and psychological distress. Although the health-related factors of kidney disease, cardiovascular disease, dyslipidemia, and hyperleptinemia are frequently reported globally, they have not been widely studied in the Pakistani population. Unlike global literature, our review could not identify any patient-related factors, such as access to health care, compliance, self-assessment and monitoring of hypertension, and a wider range of comorbidities associated with hypertension.

The use of unrestricted salt and gender were two factors that followed opposite trends in the Pakistani population as compared to the global literature. After pooling the evidence, we observed that gender was not significantly associated with hypertension and the use of unrestricted salt was protective against hypertension (unlike global studies). This might be because of the poor quality of studies and improper assessment of these variables. Another reason might be that most of the studies were cross-sectional; therefore, hypertensive individuals consuming excessive salt may still have controlled BP due to medication. Additionally, the variable "unrestricted use of salt" does not provide clear information on the actual amount of salt intake, which means that the salt used by the subjects of previous studies might still have been within limits. Considering these facts, we believe that these two variables along with the others stated above, need further investigation in the Pakistani population in future studies.

To our knowledge, this is the first well-conducted systematic review of the factors associated with hypertension specifically in Pakistan, and globally in general. In this review, we also included meta-analyses using a random-effect model to pool the effect estimates of several frequently reported factors and separately pool the estimates of factors from high-quality studies. By doing this, we were able to identify a broader range of variables using robust systematic

review methods. We assessed the quality of the included studies using the MNOQS [26,78], and only included studies where the BP was measured by a trained person and the JNC-VII guidelines [67] were used to diagnose hypertension.

## Limitations

There are several limitations to this review. Firstly, we included only observational studies, the majority (65.0%) of which had poor quality due to varying sample sizes, a lack of adjustment for potential confounders in the design and statistical analysis, no description of the sampling strategy, and the improper use of statistical methods. A few studies [39,46,51,63] were not specifically conducted to assess factors associated with hypertension, some published studies were not available even after we contacted the corresponding authors or journals, and some did not report proper statistics for the association or had errors in the reported statistics. Despite these inconsistencies, we have extracted relevant data from descriptive studies and corrected erroneous results wherever applicable.

Secondly, we could not conduct meta-analyses for several less-reported factors that were identified by only one or two studies in this review. The meta-analyses for other factors produced pooled effect estimates that provided information about the strength of associations and heterogeneity among the studies. For example, following our meta-analysis of gender, we observed that its effect on hypertension was not statistically significant and that there was a high level of heterogeneity regarding gender among the studies. A majority of the factors (65.0%) that underwent meta-analysis showed a high level of heterogeneity, but we also used random-effects models with inverse-variance weighted methods to further account for the heterogeneity. In the results section, we have provided a brief description of the sources of heterogeneity among the included studies, which can be explored in the future using meta-regression when enough studies are available. Further, the definition of hypertension varied among different studies, as did the coding or thresholds used to categorize explanatory variables. This further contributed to heterogeneity among the results.

Thirdly, all the included studies only considered a linear effect of the explanatory variables on hypertension, either through logistic regression or through other methods, such as correlations. However, it is important to recognize that some variables can have non-linear effects [79]. For example, essential dietary ingredients, such as salt and calcium, can be insufficient when taken in very small quantities but are harmful when the intake is higher than recommended. In such cases, the estimated ORs will not truly represent the relationship between the variable and hypertension.

Lastly, the variables included in this review may suffer from reporting bias because some studies may have omitted the results of variables having a non-significant association with hypertension. This could lead to an inflated value of the aggregate OR and indicate a false significant association in the meta-analysis. While we attempted to reduce the publication bias by using statistical techniques such as funnel plots and the Eggers test, it was difficult to form an opinion on the same because the number of studies was small and their analytical methods varied widely [80]. For example, some studies included additional covariates in multiple logistic regression analysis, in which case the estimated standard error might be lower than what would be obtained from a simple logistic regression analysis. In light of the above-mentioned limitations, the results of this review should be carefully interpreted when they are used.

This review is based on studies from one economically underdeveloped country, and we believe that there is much scope for this work to be extended to other countries worldwide, which can facilitate global generalizability of these findings. In this respect, our review can be considered as a protocol for further comprehensive systematic reviews. Further investigations

are required to clarify inconsistent results (e.g., the effect of gender not being significant or the protective effect of the excessive use of salt) and to re-examine the factors from a few good-quality and several poor-quality studies. In this section and the results section, we have provided a list of factors that were reported by only one or two studies, which can be investigated in further rigorously conducted research.

## Conclusion

Our study identified several socio-demographic, lifestyle, health-related, and psychological factors that were significantly (positively and negatively) associated with hypertension. These findings may help public health workers to identify high-risk groups of patients and adopt appropriate prevention and treatment strategies. We believe that in order to prioritize medical care for hypertensive patients in Pakistan, further studies are needed to investigate the direct and indirect causal inferences that link certain common factors with hypertension. Our results also highlight the poor quality of the existing evidence, particularly on hypertension and generally on overall public health research and policy in Pakistan. There is no proper national healthcare accreditation system, a lack of evidence-based guidelines for public health practices, and no enforcement of strict policies on public health, patient safety, misuse of drugs, and accountability of health care professionals and institutions. In general, no guidelines or resources outline good clinical practice and public health research in Pakistan. We strongly recommend that the Pakistani public health department and local government bodies take drastic initiatives to promote a culture of public health, involve medical doctors in research, and promote evidence-based clinical practices.

## Supporting information

**S1 Fig. Funnel plots assessing publication bias in the results for age-groups.**
(DOCX)

**S2 Fig. Forest plot from the meta-analysis (using random effect model) of gender.**
(DOCX)

**S3 Fig. Funnel plots assessing publication bias in the results for gender.**
(DOCX)

**S4 Fig. Forest plot from the meta-analysis of marital status.**
(DOCX)

**S5 Fig. Funnel plots assessing publication bias in the results for marital status.**
(DOCX)

**S6 Fig. Forest plot from the meta-analysis of the levels of education.**
(DOCX)

**S7 Fig. Funnel plots assessing publication bias in the results for the level of education.**
(DOCX)

**S8 Fig. Forest plot from the meta-analysis of income status.**
(DOCX)

**S9 Fig. Funnel plots assessing publication bias in the results for income status.**
(DOCX)

**S10 Fig. Forest plot from the meta-analysis of physical activity (active versus sedentary).**
(DOCX)

**S11 Fig. Funnel plots assessing publication bias in the results for physical activity.**
(DOCX)

**S12 Fig. Forest plot from the meta-analysis of unrestricted salt use.**
(DOCX)

**S13 Fig. Funnel plots assessing publication bias in the results for unrestricted use of salt in the food.**
(DOCX)

**S14 Fig. Forest plot from the meta-analysis of tobacco use.**
(DOCX)

**S15 Fig. Funnel plots assessing publication bias in the results for tobacco use.**
(DOCX)

**S16 Fig. Forest plot from the meta-analysis of family history of hypertension.**
(DOCX)

**S17 Fig. Funnel plots assessing publication bias in the results for family history of hypertension.**
(DOCX)

**S18 Fig. Forest plot from the meta-analysis body mass index (BMI) groups.**
(DOCX)

**S19 Fig. Funnel plots assessing publication bias in the results for BMI (overweight and obese).**
(DOCX)

**S20 Fig. Forest plot from the meta-analysis having diabetes.**
(DOCX)

**S21 Fig. Funnel plots assessing publication bias in the results for having diabetes.**
(DOCX)

**S22 Fig. Forest plot from the meta-analysis having anxiety.**
(DOCX)

**S23 Fig. Funnel plots assessing publication bias in the results for having anxiety.**
(DOCX)

**S24 Fig. Forest plot from the meta-analysis having stress.**
(DOCX)

**S25 Fig. Funnel plots assessing publication bias in the results for having stress.**
(DOCX)

**S26 Fig. Forest plot from the meta-analysis for anger-in (& -control).**
(DOCX)

**S27 Fig. Funnel plots assessing publication bias in the results for anger-in (& -control).**
(DOCX)

**S28 Fig. Forest plots for the meta-analyses of factors with effect estimates from three or more available high-quality studies (NOQAS sore $\geq$ 7).**
(DOCX)

**S1 Table. Characteristics of studies included in the systematic review.**
(DOCX)

**S2 Table. Assessing quality of the cross-sectional studies using MNOQAS.**
(DOCX)

**S3 Table. Assessing quality of case-control studies using NOQAS.**
(DOCX)

**S4 Table. PRISMA 2009 checklist.**
(DOCX)

**S5 Table. Electronic search syntaxes.**
(DOCX)

## Acknowledgments

We would like to thank Editage (www.editage.com) for English language editing.

## Author Contributions

**Conceptualization:** Muhammad Riaz, Ghazala Shah, Muhammad Asif.

**Data curation:** Muhammad Riaz, Ghazala Shah, Muhammad Asif.

**Formal analysis:** Muhammad Riaz.

**Investigation:** Muhammad Riaz, Ghazala Shah, Muhammad Asif, Kaustubh Adhikari.

**Methodology:** Muhammad Riaz.

**Project administration:** Muhammad Riaz, Ghazala Shah.

**Resources:** Muhammad Riaz, Ghazala Shah, Muhammad Asif, Asma Shah, Amani Abu-Shaheen.

**Software:** Muhammad Riaz.

**Supervision:** Muhammad Riaz.

**Validation:** Muhammad Riaz, Ghazala Shah, Muhammad Asif, Asma Shah, Kaustubh Adhikari, Amani Abu-Shaheen.

**Visualization:** Muhammad Riaz.

**Writing – original draft:** Muhammad Riaz, Ghazala Shah.

**Writing – review & editing:** Muhammad Riaz, Ghazala Shah, Muhammad Asif, Asma Shah, Kaustubh Adhikari, Amani Abu-Shaheen.

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
