## [Editor Report · Decision Letter 0]

11 Dec 2020

PONE-D-20-28653

Factors Associated with Hypertension in Pakistan: A Systematic Review and Meta-Analysis

PLOS ONE

Dear Dr, Muhammad Riaz:

Thank you for submitting your manuscript to PLOS ONE. After careful consideration, we feel that it has merit but does not fully meet PLOS ONE’s publication criteria as it currently stands. Therefore, we invite you to submit a revised version of the manuscript that addresses the points raised during the review process.

Please respond to the comments I have made below. 

A rebuttal letter that responds to each point raised by the academic editor. You should upload this letter as a separate file labeled 'Response to Reviewers'.A marked-up copy of your manuscript that highlights changes made to the original version. You should upload this as a separate file labeled 'Revised Manuscript with Track Changes'.An unmarked version of your revised paper without tracked changes. You should upload this as a separate file labeled 'Manuscript'.

We look forward to receiving your revised manuscript.

Kind regards,

James M Wright

Academic Editor

PLOS ONE

Journal Requirements:

3. Please upload the completed PRISMA checklist as a separate file.

Additional Editor Comments (if provided):

I am persuaded by the authors that this extensive amount of work deserves to be published and it is in keeping with the principals and mission of PLoSONE. There are still problems with English and the manuscript needs to be sent out for copy-ediiting for English.

Otherwise since the abstract is going to be what is most likely to be read it should be revised in the following way. Please divide the factors into those positively associated with hypertension and those that are negatively associated. The Conclusion can be changed to "Several socio-demographic, lifestyle, health-related, and psychological factors were found to be significantly positively and negatively associated with hypertension."
---

## [Author Response · Author response to Decision Letter 0]

29 Dec 2020

Response to Reviewers

Manuscript ID: PONE-D-20-28653

Manuscript title: Associated with Hypertension in Pakistan: A Systematic Review and Meta-Analysis

Dear Editor, 

We thank PLOS ONE for providing an opportunity to address the reviewers’ comments, revise and resubmit our manuscript. We thank you and the reviewers for taking time to review our manuscript and suggest the changes required. As recommended by PLOS ONE, we have used the services of Editage (www.editage.com) for English language editing. This has improved the flow and the English language of the manuscript considerably. 

In addition to this response letter to yours comments, we upload the following: (i) a revised manuscript with track changes, (ii) a clean version of our revised manuscript without tracked changes, (iii) a cover letter to the editor, (iv) a certificate issued by the Editage for English language editing, (v) three word files for figures 1-3 to be added to the main manuscript, and (vi) six supplementary word files as supporting information (in a zipped folder).

We look forward to hearing from you if there is anything else which could improve the manuscript.

Kind Regards

Comments and Responses

Comment 1

Reply 1

Thank you, we have made the changes and our updated manuscript is according to the PLOS ONE's style templates.

Comment 2

Reply 2

Thank you, we have added captions and short legends for the supporting information under the heading “Supporting Information” after the references. In addition, we now added the supporting information in 6 word files (supplementary). 

Comment 3

3. Please upload the completed PRISMA checklist as a separate file.

Reply 3

We have uploaded PRISMA checklist in a separate file as a supporting information.

Additional Editor Comments

Comment 4

I am persuaded by the authors that this extensive amount of work deserves to be published and it is in keeping with the principals and mission of PLoSONE. There are still problems with English and the manuscript needs to be sent out for copy-ediiting for English.

Reply 4

We thank the editor for valued comments, we agree that our manuscript required copy-editing for English language. We have used the services of Editage (www.editage.com) for English language editing. Their certificate is upload in a separate file.

Comment 5

Otherwise since the abstract is going to be what is most likely to be read it should be revised in the following way. Please divide the factors into those positively associated with hypertension and those that are negatively associated. The Conclusion can be changed to "Several socio-demographic, lifestyle, health-related, and psychological factors were found to be significantly positively and negatively associated with hypertension."

Reply 5

In addition to the above suggestions, the English language editor has suggested some further changes to the abstract. We have revised the abstract to reflect the suggested changes. Also, we have made changes to the conclusion as suggested above.

---

## [Editor Report · Decision Letter 1]

13 Jan 2021

Factors associated with hypertension in Pakistan: a systematic review and meta-analysis

PONE-D-20-28653R1

Dear Dr. Muhammad Riaz,

We’re pleased to inform you that your manuscript has been judged scientifically suitable for publication and will be formally accepted for publication once it meets all outstanding technical requirements.

Kind regards,

James M Wright

Academic Editor

PLOS ONE
---

## [Editor Report · Acceptance letter]

20 Jan 2021

PONE-D-20-28653R1 

Factors associated with hypertension in Pakistan: a systematic review and meta-analysis 

Dear Dr. Riaz:

I'm pleased to inform you that your manuscript has been deemed suitable for publication in PLOS ONE. Congratulations! Your manuscript is now with our production department. 

Kind regards, 

on behalf of

Professor James M Wright 

Academic Editor

PLOS ONE